# Age-Related Hearing Loss in C57BL/6J Mice Is Associated with Mitophagy Impairment in the Central Auditory System

**DOI:** 10.3390/ijms21197202

**Published:** 2020-09-29

**Authors:** Cha Kyung Youn, Yonghyun Jun, Eu-Ri Jo, Sung Il Cho

**Affiliations:** 1Department of Premedical Science, Chosun University College of Medicine, Gwangju 61452, Korea; threefold@hanmail.net; 2Department of Anatomy, Chosun University College of Medicine, Gwangju 61452, Korea; jyh1483@chosun.ac.kr; 3Department of Otolaryngology-Head and Neck Surgery, Chosun University College of Medicine, Gwangju 61452, Korea; maria2046@hanmail.net

**Keywords:** auditory cortex, presbycusis, age-related hearing loss, mitochondria, mitophagy

## Abstract

Aging is associated with functional and morphological changes in the sensory organs, including the auditory system. Mitophagy, a process that regulates the turnover of dysfunctional mitochondria, is impaired with aging. This study aimed to investigate the effect of aging on mitophagy in the central auditory system using an age-related hearing loss mouse model. C57BL/6J mice were divided into the following four groups based on age: 1-, 6-, 12-, and 18-month groups. The hearing ability was evaluated by measuring the auditory brainstem response (ABR) thresholds. The mitochondrial DNA damage level and the expression of mitophagy-related genes, and proteins were investigated by real-time polymerase chain reaction and Western blot analyses. The colocalization of mitophagosomes and lysosomes in the mouse auditory cortex and inferior colliculus was analyzed by immunofluorescence analysis. The expression of genes involved in mitophagy, such as *PINK1*, *Parkin*, and *BNIP3* in the mouse auditory cortex and inferior colliculus, was investigated by immunohistochemical staining. The ABR threshold increased with aging. In addition to the mitochondrial DNA integrity, the mRNA levels of *PINK1*, *Parkin*, *NIX*, and *BNIP3*, as well as the protein levels of *PINK1*, *Parkin*, *BNIP3*, *COX4*, LC3B, mitochondrial oxidative phosphorylation (OXPHOS) subunits I–IV in the mouse auditory cortex significantly decreased with aging. The immunofluorescence analysis revealed that the colocalization of mitophagosomes and lysosomes in the mouse auditory cortex and inferior colliculus decreased with aging. The immunohistochemical analysis revealed that the expression of *PINK1*, *Parkin*, and *BNIP3* decreased in the mouse auditory cortex and inferior colliculus with aging. These findings indicate that aging-associated impaired mitophagy may contribute to the cellular changes observed in an aged central auditory system, which result in age-related hearing loss. Thus, the induction of mitophagy can be a potential therapeutic strategy for age-related hearing loss.

## 1. Introduction

Presbycusis or age-related hearing loss is the most common type of sensory disorder among the elderly that results in irreversible sensorineural hearing loss. In the elderly, hearing loss leads to various psychophysical problems, including poor communication skills [1]. The major site of auditory system that is affected by age-related changes is the cochlea. Age-related pathological changes in the cochlea include the loss of hair cells and spiral ganglion neurons, as well as atrophy of stria vascularis [2,3]. However, sound is perceived through the whole auditory system from the ear to the auditory cortex. The age-related changes affect the entire auditory system, including the peripheral and central auditory systems [3]. Previous studies have reported age-related changes in the superior olivary complex, inferior colliculus, and auditory cortex of the central auditory system [4,5,6]. The functional and morphological changes associated with aging include cortical atrophy and neural pathologies, such as decreased number of dendrites, synapses, and neuronal fibers [7]. The central auditory system dysfunction results in defective processing of acoustical information and limited comprehension of speech [8]. Mitochondria are important for the production of adenosine triphosphate (ATP) and regulation of cellular homeostasis. However, mitochondria are also a major site for reactive oxygen species (ROS) production, which promotes oxidative damage. The frequency of mitochondrial DNA (mtDNA) mutations increases exponentially with age, which promotes mitochondrial dysfunction and contributes to cellular aging and age-related diseases [9]. The metabolic and self-repair abilities of cells progressively decline with aging, which leads to accumulations of dysfunctional or damaged organelles and proteins [10]. The accumulation of dysfunctional mitochondria results in impaired energy metabolism, which subsequently plays a key role in the development of age-related hearing loss. Human mitochondrial diseases are associated with hearing loss. Additionally, aging is associated with increased number of mitochondrial defects [11]. These findings indicate that mitochondrial dysfunction is involved in age-related hearing loss.

Autophagy is a cellular process that degrades and recycles the damaged or superfluous organelles. In addition to the removal of damaged organelles, autophagy aids in clearing misfolded proteins within the cells. This indicates that autophagy is one of the major mechanisms that contribute to cell survival during aging [12]. Mitochondrial autophagy (mitophagy) is a specialized quality control mechanism that maintains mitochondrial homeostasis by removing dysfunctional mitochondria from the cell [13]. Various studies have reported the beneficial roles of autophagy in the auditory system. The induction of mitophagy mitigated the carbonyl cyanide m-chlorophenyl hydrazone (CCCP)-induced cytotoxicity in the auditory cell line [14]. Additionally, the upregulation of autophagy has been reported to attenuate the damage to the inner ear and prevent hearing loss [10].

Unlike the peripheral auditory system, the detailed mechanism underlying age-related hearing loss in the central auditory system has not been elucidated. This study hypothesized that impaired mitophagy might be the underlying mechanism of age-related central auditory system dysfunction, which leads to age-related hearing loss. This study aimed to investigate the correlation between mitophagy impairment and age-related changes of the central auditory system.

## 2. Results

### 2.1. Hearing Thresholds and mtDNA Damage in the Auditory Cortex Increase with Aging

To investigate the change of hearing loss with aging, hearing thresholds of 8, 16, and 32 kHz in 1-, 6-, 12-, and 18-month groups were evaluated by measuring auditory brainstem response (ABR). The ABR thresholds of 1-, 6-, 12-, and 18-month groups at a frequency of 8 kHz were 36 ± 2.5 dB, 54 ± 2.5 dB, 64 ± 2.5 dB, and 78 ± 2.0 dB SPL, respectively. The ABR thresholds of 1-, 6-, 12-, and 18-month groups at a frequency of 16 kHz were 24 ± 2.5 dB, 50 ± 4.5 dB, 60 ± 3.2 dB, and 84 ± 2.5 dB SPL, respectively. The ABR thresholds of 1-, 6-, 12-, and 18-month groups at a frequency of 32 kHz were 34 ± 2.5 dB, 64 ± 2.5 dB, 72 ± 3.7 dB, and 82 ± 3.7 dB SPL, respectively. The ABR thresholds at all frequencies progressively increased with aging (*n* = 15, Figure 1A). To examine the mtDNA integrity in the auditory cortex, the ratio of long mtDNA to short mtDNA was investigated in the 1-, 6-, 12-, and 18-month groups. The ratio of the mtDNA integrity significantly decreased with aging (*n* = 5, *p* < 0.05, Figure 1B). These results indicate that mitochondrial damage in the auditory cortex and hearing thresholds increased with aging.

### 2.2. Mitophagy-Related Genes and Proteins in the Auditory Cortex Are Downregulated with Aging

The effect of aging on mitophagy in the mouse auditory cortex that is composed of tightly packed neurons and glial cells was evaluated by measuring the expression levels of genes and proteins associated with mitophagy in the 1-, 6-, 12-, and 18-month groups. The mRNA levels of *PINK1*, *Parkin*, *BNIP3*, and *NIX* in the mouse auditory cortex significantly decreased with aging (*n* = 5, *p* < 0.05, Figure 2A). Additionally, the protein expression levels of *PINK1*, *Parkin*, and *BNIP3* in the mouse auditory cortex significantly decreased with aging. The protein expression levels of *COX4* (a mitochondrial marker) and LC3B (an autophagy marker) in the mouse auditory cortex significantly decreased with aging (*n* = 5, *p* < 0.01, Figure 2B,C). The mitochondrial function was investigated by analyzing the expression levels of mitochondrial oxidative phosphorylation (OXPHOS) complex. The OXPHOS subunits I–IV are components of the electron transport chain that generate energy for subunit V (ATP synthase) to produce ATP. The expression levels of OXPHOS subunits I–IV in the mouse auditory cortex significantly decreased with aging. However, the expression level of OXPHOS subunit V non-significantly decreased with aging (*n* = 5, *p* < 0.01, Figure 2B,C). These result indicated that the expression of mitophagy-related genes and proteins, as well as the mitochondrial function for energy generation, decreased with aging in the mouse auditory cortex.

### 2.3. Mitophagy Is Impaired in the Central Auditory System with Aging

To examine the change in mitophagosomes and mitophagolysosomes in the aged auditory cortex and inferior colliculus, immunofluorescence analysis was performed. The formation of mitophagosomes was examined by analyzing the colocalization of LC3B and TOM20, a major mitochondrial receptor. The colocalization of LC3B and TOM20 in the mouse auditory cortex and inferior colliculus significantly decreased with aging (*n* = 5, *p* < 0.05, Figure 3A and Figure 4A). The formation of mitophagolysosomes was investigated by analyzing the colocalization of LAMP1, a marker of lysosome, and TOM20 by immunofluorescence analysis. The colocalization of LAMP1 and TOM20 in the auditory cortex and inferior colliculus significantly decreased with aging (*n* = 5, *p* < 0.05, Figure 3B and Figure 4B). The effect of age on major mitophagy-related pathways was analyzed by examining the protein expression levels of *PINK1*, *Parkin*, and *BNIP3* by IHC staining. The IHC scores of *PINK1*, *Parkin*, and *BNIP3* in the mouse auditory cortex and inferior colliculus significantly decreased with aging (*n* = 5, *p* < 0.05, Figure 5 and Figure 6). These results indicated that mitophagy progressively decreased with aging in the mouse auditory cortex and inferior colliculus.

## 3. Discussion

The quality of hearing is dependent on the integrity of peripheral auditory system and central auditory system. Cochlea generates auditory nerve impulses that travel through the superior olivary complex and inferior colliculus. Next, the impulses undergo final integration within the auditory cortex. The inferior colliculus, which is important for sound localization, is the relay station on the ascending auditory pathway. The auditory cortex, which integrates auditory information, is necessary for perception and interpretation of the auditory stimulus [15]. Age-related hearing loss is associated with changes in the peripheral auditory system and central auditory system. The central sound processing deficits in the elderly prevent processing of speech and other acoustic signals in noisy or complex environments [1]. ATP is generated in the cells by mitochondria, which are known as the powerhouse of the cell, through the OXPHOS system. The energy supply is critical for the ATP-dependent neurotransmission in the nervous system, including the auditory system [16]. The function and morphology of the mitochondria decline with aging, which results in an energy deficit and age-dependent decline in organ function [17]. In this study, the expression of OXPHOS subunits I–IV was downregulated with aging, which indicated age-dependent decline of mitochondrial function.

Damaged mitochondria trigger the leakage of electrons and the production of harmful ROS, which can damage proteins, membrane lipids, and nucleic acids [18]. Age-associated mitochondrial dysfunctions are compensated by mitochondrial quality control mechanisms, which mediate the balance between mitochondrial biogenesis and efficient removal of damaged mitochondria. In autophagy, the damaged organelles and protein aggregates fuse with the lysosome and undergo degradation [19]. Autophagy may have a dual role. Early upregulation of autophagy may have a pro-survival role and serve to remove damaged mitochondria; however, if this pro-survival attempt fails, autophagy may trigger cell death pathways [20]. Mitophagy, a specialized form of autophagy, maintains a healthy mitochondrial population by regulating the turnover of damaged mitochondria [21]. During mitophagy, damaged mitochondria are engulfed by the autophagosome, which fuses with lysosome to form mitophagolysosome in which the mitochondria is degraded. Mitophagy impairment causes progressive accumulation of defective mitochondria, which results in cell and tissue damages [22]. In this study, aging was associated with progressive decrease in the formation of mitophagosomes and mitophagolysosomes in the mouse auditory cortex and inferior colliculus, which indicated that mitophagy impairment was age-dependent.

The major pathway for mitochondrial quality control in mammalian cells is the *PINK1*/*Parkin* pathway. *PINK1* localizes to the outer mitochondrial membrane and recruits *Parkin*, an E3 ubiquitin ligase, upon loss of the mitochondrial membrane potential. *Parkin* promotes ubiquitination of mitochondrial outer membrane proteins and subsequently activates the ubiquitin–proteasome system. This promotes the engulfment of impaired mitochondria by phagophore and formation of mitophagosomes [23]. The *PINK1*/*Parkin* pathway also interferes with other mitochondrial quality control mechanisms, such as mitochondrial dynamics and mitochondria-derived vesicles to maintain energy homeostasis [24]. This study demonstrated that aging was associated with downregulation of *PINK1*/*Parkin* pathway in the mouse auditory cortex and inferior colliculus. This indicated that *PINK1*/*Parkin* pathway-dependent mitophagy was impaired with aging to maintaining mitochondrial homeostasis in the central auditory system.

In contrast to *PINK1*/*Parkin*-mediated mitophagy, which requires the translocation of *Parkin*, the *Parkin*-independent mitophagy regulators are localized at the mitochondrial outer membrane. *BNIP3* and its homolog *NIX*/*BNIP3L* are key regulators involved in the *Parkin*-independent regulation of mitophagy [25]. These proteins, which function as adaptor proteins, bind to LC3 on autophagosomes and promote the autophagic engulfment of mitochondria [26]. *BNIP3* and *NIX*, which are upregulated during neuronal stress, are involved in stress relief [27]. This study demonstrated that the expression of *BNIP3*, a *Parkin*-independent mitophagy regulator, was downregulated in the mouse auditory cortex and inferior colliculus with aging. This indicated that *Parkin*-independent mitophagy, which mitigates neuronal stress, was also downregulated in the central auditory system with aging.

Mitophagy appears to play an important role in cochlear hair cell loss and hearing deterioration during aging. SIRT1 is an NAD-dependent deacetylase that modulates mitophagy. A previous study has shown that activation of SIRT1 increased the levels of *PINK1* and *Parkin* in old mice and attenuated age-related cochlear hair cell loss [28]. Moreover, reduced mitophagy was observed in aged cochlea, which increased hearing loss [29]. This is supported by a study found that the inhibition of dynamin-related protein-1 (DRP-1), which initiates mitophagy, promotes cochlear hair cell senescence and exacerbates age-related hearing loss [30].

Mitophagy serves as a major quality control mechanism to remove aged and impaired mitochondria in neurons [16]. The disruption of mitophagy has been associated with various neurodegenerative disorders, including Alzheimer’s, Parkinson’s, and Huntington’s diseases. Consistent with the critical role of mitophagy, the accumulation of dysfunctional mitochondria has been reported in the brain of patients with Parkinson’s disease and hippocampal neurons of a mouse Alzheimer disease model [31,32,33,34]. The upregulation of *PINK1*/*Parkin* axis, which is associated with early onset Parkinson’s disease, has been reported to exert neuroprotective effects [35,36]. The downregulated expression of *Parkin* and abnormal *PINK1* accumulation were reported in the brains of patients with Alzheimer’s disease [37]. The dentate gyrus of Huntington’s disease mouse model exhibited decreased levels of mitophagy [38]. This study demonstrated that mitophagy and mitochondrial function are impaired in the central auditory system of age-related hearing loss mouse model. This suggests that the maintenance of mitochondrial integrity is important in the central auditory system and that enhancing the clearance of impaired mitochondria through the induction of mitophagy could be a promising strategy for preventing age-related hearing loss.

## 4. Materials and Methods

### 4.1. Mice and Animal Care

C57BL/6J mice were purchased from Animal Facility of Aging Science, KBSI Gwangju Center (Gwangju, South Korea). The animal experiments were approved by the Chosun University Institutional Animal Care and Use Committee (approval No. CIACUC2018-A0045; approval date: Jan 15 2019). All experiments were performed in accordance with relevant guidelines and strict regulations approved by the committee. The mice were randomly divided into four groups (*n* = 15 per group) based on age: 1-, 6-, 12-, and 18-month groups. The mice were subjected to hearing tests and euthanized by cervical dislocation under 5% isoflurane anesthesia. The mouse brain was excised for further studies.

### 4.2. Auditory Brainstem Response (ABR) Test

The mice (*n* = 15 per group) were anesthetized by injecting ketamine (50 mg/mL, Yuhan, Seoul, Korea) and xylazine (23.32 mg/mL, Bayer Korea, Seoul, Korea). The ketamine/xylazine cocktail was prepared by mixing 3.5 mL of ketamine, 1 mL of xylazine, and 8 mL of sterile water. The cocktail was injected intraperitoneally at a dose of 0.1 mL/20 g bodyweight. The hearing tests were conducted in a soundproof chamber. The body temperature of mice was maintained at 37–38 °C by placing the anesthetized mice on a heating pad and monitored using a rectal probe throughout the recording period. The ABR thresholds in both ears were recorded from the scalp of the mice using the computerized Intelligent Hearing System (IHS, Miami, FL, USA) equipped with the Smart-EP software. Subcutaneous needle electrode were inserted on the vertex and overlying the ventral region of the left and right bullae. The tone burst stimuli were generated in waveforms with 256 stimuli (rate: 21.1/s) at frequencies of 8 kHz, 16 kHz, and 32 kHz. The ABR waveforms were recorded on a sampling period of 10 ms from 10–80 dB sound pressure level (SPL) intervals below the maximum amplitude. The ABR threshold was defined as the lowest stimulus level at which response peaks for waves were visible in the evoked trace.

### 4.3. Quantitative Real-Time Polymerase Chain Reaction (qRT-PCR)

The mice (*n* = 5 per group) were sacrificed and both sides of the auditory cortex from each mouse were rapidly excised. To analyze the mtDNA integrity, total DNA was isolated from the auditory cortex tissues using the Accuprep Genomic DNA extraction Kit (Bioneer, Daejeon, Korea). For PCR amplification of mtDNA, 20 ng of the total DNA was used. The long fragment of mtDNA (10.1 kb, *Mus musculus*) was amplified using the following primers: 5′-GCCAGCCTGACCCATAGCCATAATAT-3′ and 5′-GAGAGATTTTATGGGTGTAATGCGG-3′. The short fragment of mtDNA (117 bp, *Mus musculus*) was amplified using the following primers: 5′-CCCAGCTACTACCATCATTCAAGT-3′ and 5′-GATGGT TTGGGAGATTGGTTGATGT-3′. The PCR conditions for long fragment amplification were as follows: initial denaturation at 95 °C for 30 s, followed by 40 cycles of 95 °C for 15 s and 60 °C for 720 s. The PCR conditions for short fragment amplification were as follows: initial denaturation at 95°C for 30 s, followed by 40 cycles of 95 °C for 15 s and 60 °C for 90 s. The qRT-PCR analysis was performed in the Applied Biosystems™ 7500 Fast Real-Time PCR System (Applied Biosystems, Foster City, CA, USA) using SYBR Premix Ex Taq^TM^ II (TaKaRa Bio, Shiga, Japan). The short fragment of mtDNA was used as the reference gene. The mtDNA integrity was determined by examining mtDNA damage by calculating the ratio of long mtDNA to short mtDNA.

To analyze the mRNA expression levels of mitophagy-related genes, total RNA was isolated from the auditory cortex tissues using Hybrid-RTM (GeneAll, Seoul, Korea), following the manufacturer’s instructions. The extracted RNA (1 μg) was reverse transcribed into cDNA using the M-MLV cDNA Synthesis Kit (Enzynomics, Daejeon, Korea). The qRT-PCR analysis was performed using the SYBR Premix Ex Taq^TM^ kit (TaKaRa Bio). The primers used for qRT-PCR analysis were as follows: *PINK1* forward, 5′-GCTTGCCAATCCCTTCTATG-3′ and *PINK1* reverse, 5′-CTCTCGCTGGAGCAGTGAC-3′; *Parkin* forward, 5′-AAACCGGATGAGTGGTGAGT-3′ and *Parkin* reverse, 5′-AGCTACCGACGTGTCCTTGT-3′; *BNIP3* forward, 5′-GCTCCCAGACACCACAAGAT-3′ and *BNIP3* reverse, 5′-TGAGAGTAGCTGTGCGCTTC-3′; *NIX* forward, 5′-GCAGGGACCATAGCTCTCAG-3′ and *NIX* reverse, 5′-TGCTCAGTCGCTTTCCAATA-3′; *GAPDH* forward, 5′-GTATTGGGCGCCTGGTCACC-3′ and *GAPDH* reverse, 5′-CGCTCCTGGAAGATGGTGATGG-3′. The *GAPDH* gene was used as the reference gene. The expression of target gene was normalized to that of reference gene. The fold change in gene expression was calculated by the 2^−ΔΔCt^ method.

### 4.4. Western Blot Analysis

The two sides of the auditory cortex were rapidly excised from the sacrificed mice (*n* = 5 per group). The tissues were homogenized using a homogenizer (Omni International, Kennesaw, GA, USA) in 300 µL lysis buffer. The homogenized samples were sonicated twice (each for 10 s) in a sonicator (Sonics & Materials Inc., Newtown, CT, USA) and then centrifuged at 13,000 rpm for 15 min to obtain soluble proteins. The proteins (20 µg) were subjected to sodium dodecyl polyacrylamide gel electrophoresis (SDS-PAGE) using a 12% gel. The resolved proteins were transferred to a polyvinylidene fluoride (PVDF) membrane (Millipore Corp, Burlington, MA, USA). The membrane was blocked with 5 % skim milk prepared in Tris buffer saline/Tween-20 (TBS-T) at room temperature for 1 h. The membrane was then washed with TBS-T and incubated with the following primary antibodies overnight at 4 °C: *PINK1*, *Parkin*, *BNIP3* (all 1:1000, ThermoFisher Scientific, Waltham, MA, USA), *COX4*, LC3B (1:1000, Cell Signaling Technology, Danvers, MA, USA), and β-actin (1:4000, Santa Cruz Biotechnology, Dallas, TX, USA) antibodies. Following the incubation, the membrane was washed three times with TBS-T (10 min/step) and incubated with the following secondary antibodies for 2 h: anti-mouse (1:4000, Jackson ImmunoResearch, West Grove, PA, USA) or anti-rabbit (1:4000, Jackson ImmunoResearch) antibodies, as appropriate. The protein bands were visualized using a Western blot detection system (Millipore, Burlington, MA, USA). The blots were analyzed using an image analyzer (FUSION Solo X, VILBER, Paris, France).

### 4.5. Immunostaining

The mice (*n* = 5 per group) were anesthetized with 5% isoflurane and perfused transcardially with 4% paraformaldehyde. Brain tissue was then carefully excised from the skull and immersed in 4% paraformaldehyde overnight. The brain tissue was dehydrated using a graded series of alcohol and embedded in paraffin wax. The paraffin-embedded brain tissue was serially sectioned into 6-μm thick sections at the level of the auditory cortex and inferior colliculus. The sections were then mounted on glass slides (Fisher Scientific, Hampton, NH, USA), and representative sections were used for experiments. 

For immunofluorescence, sections of the auditory cortex and inferior colliculus region were blocked in 0.5% bovine serum albumin (BSA) solution. The sections were incubated with primary antibodies against translocase of the outer membrane 20 (TOM20) (1:200; Santa Cruz Biotechnology), LC3B (1:200; Cell Signaling Technology), and LAMP1 (1:200, Santa Cruz Biotechnology) overnight at 4 °C. The sections were then incubated with the following fluorescein isothiocyanate (FITC)-conjugated secondary antibodies: chicken anti-rabbit Alexa Fluor 488 and chicken anti-mouse Alexa Fluor 594 (1:200, Invitrogen, Carlsbad, CA, USA). Nuclei were counterstained with 4,6-diamidino-2-phenylindole (DAPI) (GBI Labs, Bothell, WA, USA). Immunofluorescence of the sections was detected using a confocal microscopy (Carl Zeiss, Oberkochen, Germany) and analyzed using Zeiss microscope image software ZEN (Carl Zeiss). Colocalization coefficients were quantified using ImageJ software (National Institutes of Health, Bethesda, MD, USA).

For immunohistochemical (IHC) analysis, five sections for each animal were deparaffinized, rehydrated, and rinsed three times with 0.1 M phosphate-buffered saline (PBS). The sections were incubated in 0.01 M sodium citrate buffer and subjected to microwave antigen retrieval. The sections were then treated with 0.3% hydrogen peroxide for 20 min to block endogenous peroxidase activity. The sections were rinsed with PBS and blocked with normal horse serum in 0.5% BSA solution for 30 min at room temperature. The sections were then incubated with the following primary antibodies: *PINK1*, *Parkin*, and *BNIP3* (all 1:100, ThermoFisher Scientific). The sections were washed three times with PBS for 10 min. The immunoreactivity was visualized using biotinylated anti-rabbit IgG, chromogen 3,3-diamino-benzidine substrate, and the avidin-biotin-peroxidase (ABC) detection system (Vectastain ABC Elite Kit, Vector Laboratories, Burlingame, CA, USA). The sections were counterstained with thionine for 30 s and dehydrated. Polymount mounting medium (Polysciences, Warrington, PA, USA) was added to the sections, which were then covered with a coverslip. The immunoreactivity of *PINK1*, *Parkin*, and *BNIP3* was scored based on the staining intensity (0, none; 1, weak; 2, moderate; 3, strong) and percent positive cells (0, none; <5%, 1; 6–25%, 2; 26–50%, 3; 51–75%, 4; >76%, 5). The results are expressed as the product of both scores.

### 4.6. Statistical Analyses

All statistical analyses were performed with the SPSS 24.0 software (SPSS Inc., Chicago, IL, USA). Comparisons between age groups were analyzed using one-way analysis of variance (ANOVA), followed by the Tukey’s honestly significant difference (HSD) post-hoc test. Data on age and the expression of mitophagy-related genes and proteins were analyzed using two-way ANOVA. The differences were considered statistically significant when the *p*-value was less than 0.05.

## Figures and Tables

**Figure 1 ijms-21-07202-f001:**
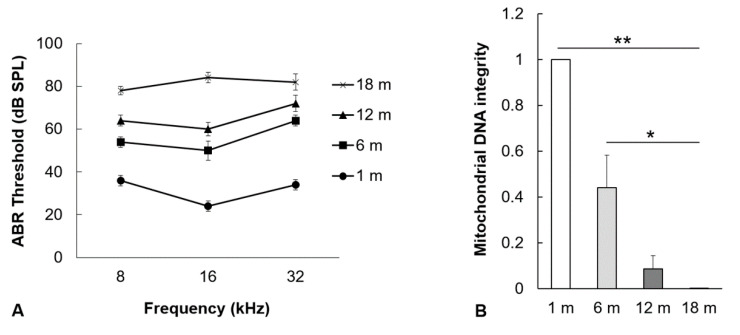
The effect of aging on auditory brainstem response (ABR) thresholds and mitochondrial DNA (mtDNA) integrity. (**A**) The hearing thresholds of mice at frequencies of 8 kHz, 16 kHz, and 32 kHz increased with aging (fifteen mice per group). (**B**) The mtDNA integrity was assessed by measuring the expression level of 10 kb amplicon. The mtDNA integrity in the mouse auditory cortex significantly decreased with aging. The data are shown as mean ± standard error of mean (five mice per group; 1 m, 1 month; 6 m, 6 months; 12 m, 12 months; 18 m, 18 months; SPL, sound pressure level). * *p* < 0.05, ** *p* < 0.01.

**Figure 2 ijms-21-07202-f002:**
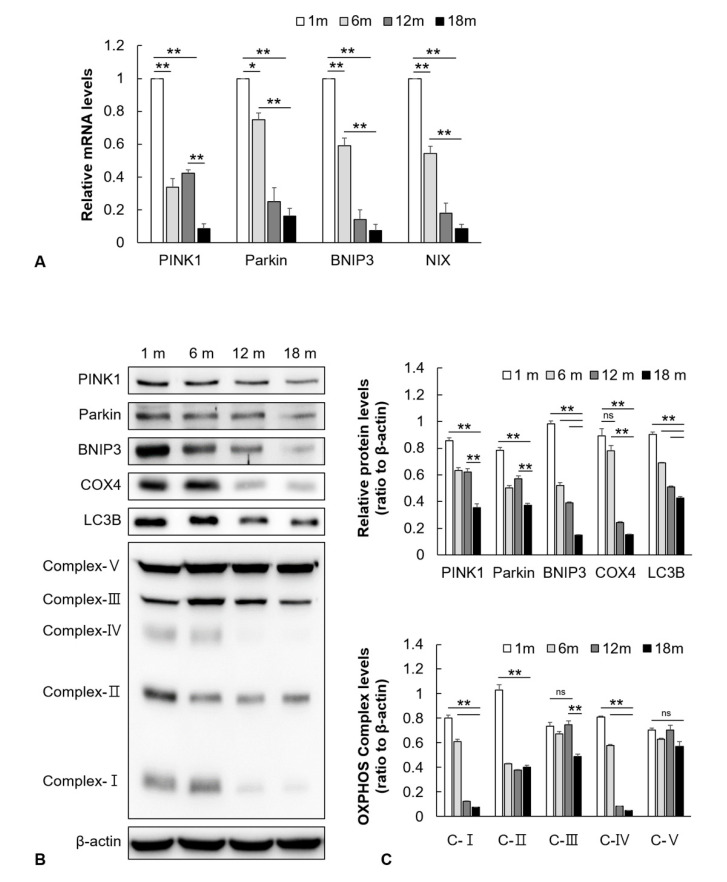
The effect of aging on the expression of mitophagy-related genes, proteins, and oxidative phosphorylation (OXPHOS) subunits in the mouse auditory cortex. (**A**) The mRNA expression levels of *PINK1*, *Parkin*, *BNIP3*, and *NIX* in the mouse auditory cortex significantly decreased with aging. (**B**) The Western blotting analysis of *PINK1*, *Parkin*, *BNIP3*, *COX4*, LC3B, and OXPHOS subunits. β-actin was used as the loading control. (**C**) The relative protein expression levels of *PINK1*, *Parkin*, *BNIP3*, *COX4*, LC3B, and OXPHOS subunits I–IV in the mouse auditory cortex significantly decreased with aging. The data are shown as mean ± standard error of mean (five mice per group; 1 m, 1 month; 6 m, 6 months; 12 m, 12 months; 18 m, 18 months). * *p* < 0.05, ** *p* < 0.01, ns: not significant.

**Figure 3 ijms-21-07202-f003:**
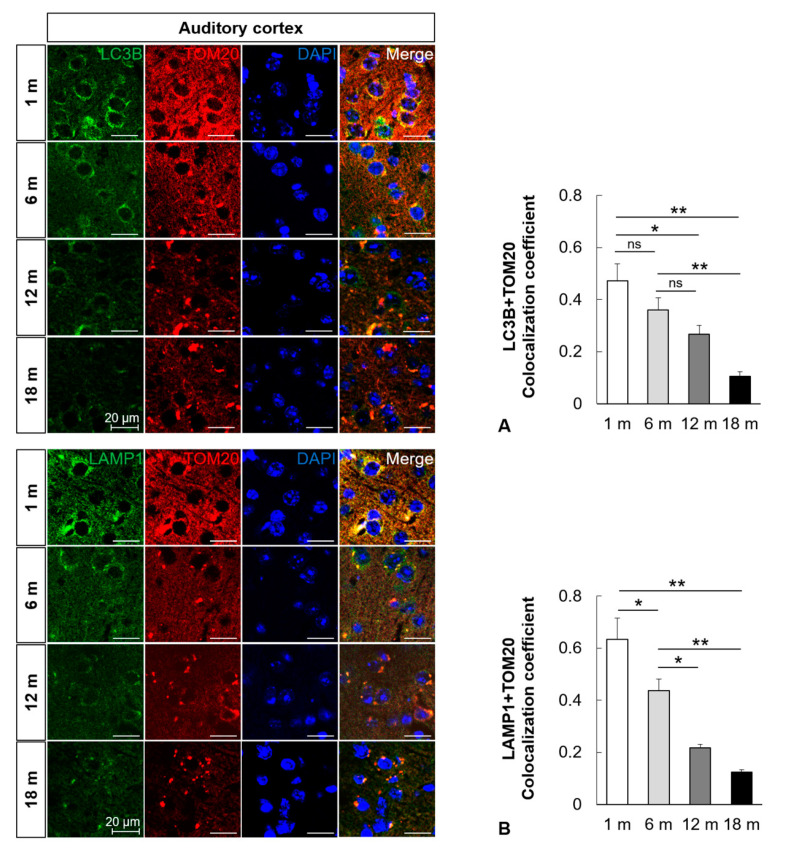
Impairment of mitophagy in the mouse auditory cortex with aging. (**A**) Colocalization analysis of autophagosomes and mitochondria. Immunofluorescence analysis revealed that the colocalization (yellow puncta, the overlap color) of LC3B (green) and TOM20 (red) in the mouse auditory cortex significantly decreased with aging. (**B**) Colocalization analysis of lysosomes and mitophagosomes. Immunofluorescence analysis revealed that the colocalization (yellow puncta, the overlap color) of LAMP1 (green) and TOM20 (red) in the mouse auditory cortex significantly decreased with aging. The data are shown as mean ± standard error of mean (five mice per group; 1 m, 1 month; 6 m, 6 months; 12 m, 12 months; 18 m, 18 months; DAPI, 4′,6-diamidino-2-phenylindole). * *p* < 0.05, ** *p* < 0.01, ns: not significant.

**Figure 4 ijms-21-07202-f004:**
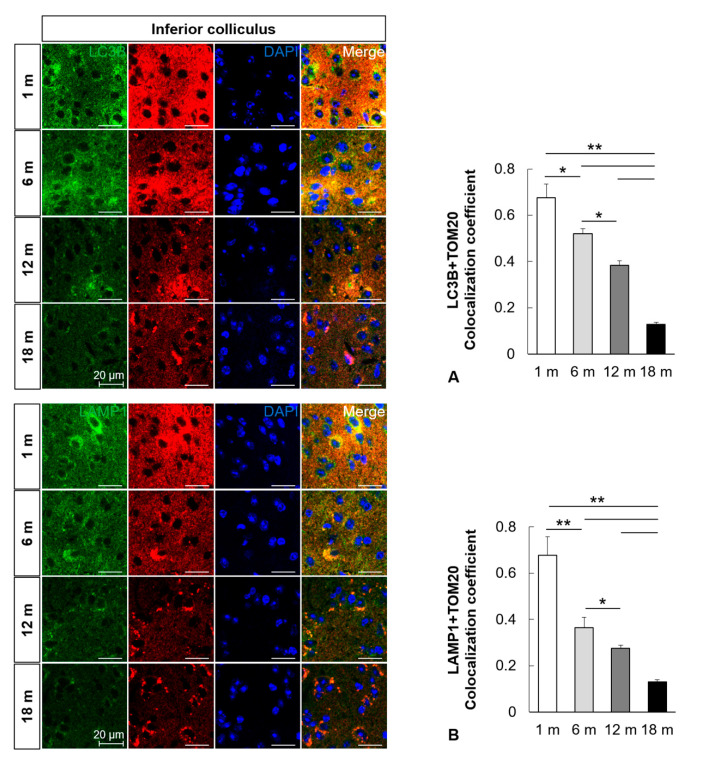
Impairment of mitophagy in the mouse inferior colliculus with aging. (**A**) Colocalization analysis of autophagosomes and mitochondria. Immunofluorescence analysis revealed that the colocalization (yellow puncta, the overlap color) of LC3B (green) and TOM20 (red) in the mouse inferior colliculus significantly decreased with aging. (**B**) Colocalization analysis of lysosomes and mitophagosomes. Immunofluorescence analysis revealed that the colocalization (yellow puncta, the overlap color) of LAMP1 (green) and TOM20 (red) in the mouse inferior colliculus significantly decreased with aging. The data are shown as mean ± standard error of mean (five mice per group; 1 m, 1 month; 6 m, 6 months; 12 m, 12 months; 18 m, 18 months; DAPI, 4′,6-diamidino-2-phenylindole). * *p* < 0.05, ** *p* < 0.01.

**Figure 5 ijms-21-07202-f005:**
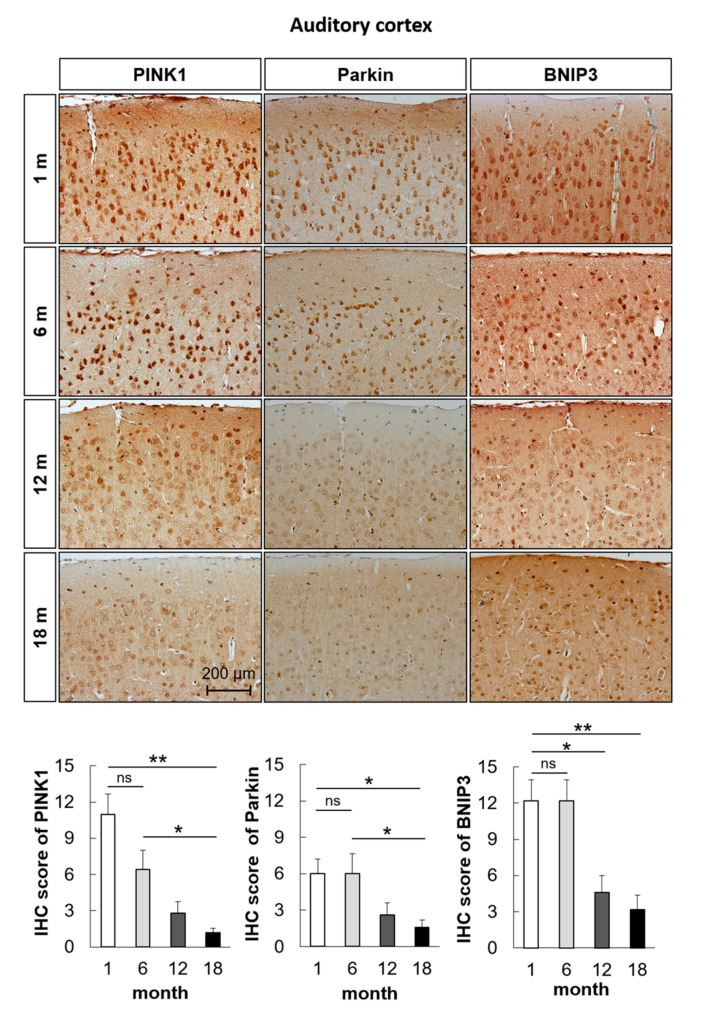
The protein expression levels of *PINK1*, *Parkin*, and *BNIP3* in the mouse auditory cortex decrease with aging. Immunohistochemical (IHC) scores of *PINK1*, *Parkin*, and *BNIP3* in the mouse auditory cortex significantly decreased with aging. The data are shown as mean ± standard error of mean (five mice per group; 1 m, 1 month; 6 m, 6 months; 12 m, 12 months; 18 m, 18 months). * *p* < 0.05, ** *p* < 0.01, ns: not significant.

**Figure 6 ijms-21-07202-f006:**
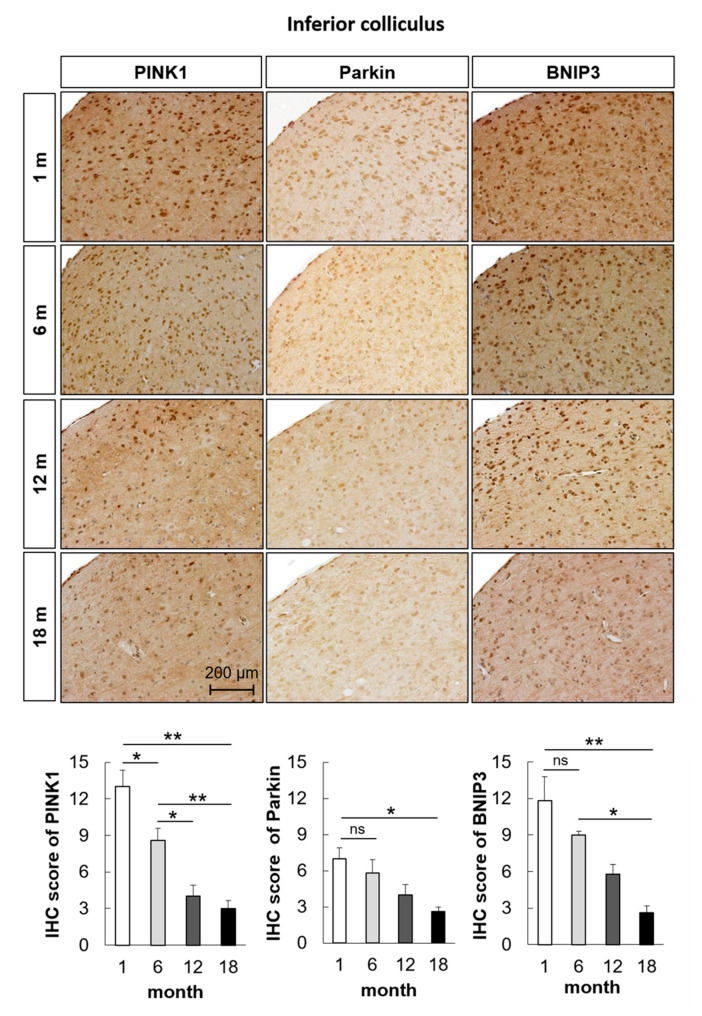
The protein expression levels of *PINK1*, *Parkin*, and *BNIP3* in the mouse inferior colliculus decrease with age. Immunohistochemical (IHC) scores of *PINK1*, *Parkin*, and *BNIP3* in the mouse inferior colliculus significantly decreased with aging. The data are shown as mean ± standard error of mean (five mice per group; 1 m, 1 month; 6 m, 6 months; 12 m, 12 months; 18 m, 18 months). * *p* < 0.05, ** *p* < 0.01, ns: not significant.

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
