# Peer review of "Age-Related Hearing Loss in C57BL/6J Mice Is Associated with Mitophagy Impairment in the Central Auditory System"

_ijms, 2020, doi:10.3390/ijms21197202_

Round 1

Reviewer 1 Report

The investigation of autophagy/mitophagy in sensorineural hearing losses is a relatively new approach. This study shows the down regulation of Parkin-dependent (PINK1/Parkin) and Parkin-independent (BNIP3/NIX) regulators of mitophagy (at the RNA and protein levels) and the decrease of mitophagosome- and mitophagolysosome formation in the mouse auditory cortex and inferior colliculus by age progression. The decline of mitoDNA integrity and expression levels of OXPHOS subunits I-IV was also detected in the auditory cortex with age.

The authors now repeated in the central auditory system their previous investigation in the cochlea with similar results and conclusions (Exp Gerontol. 2020 Aug;137:110946). The study is essentially correlative and lacks the investigation of causative relationships.

Comments and questions:

1.) The authors have published a very similar study in the cochlea (Exp Gerontol. 2020 Aug;137:110946). This paper should be cited in this one.

2.) Autophagy may have dual-role (Janus face; e.g. Antioxid Redox Signal 2012, 16:263). It may be favourable to mention and explain this in the Discussion.

3.) The level of auto/mitophagy related RNAs/proteins decreased with age. What could be the cellular location of these decreases? Neurons or Glial cells?

4.) One-way ANOVA is appropriate for testing the effect of one independent variable (age) on the dependent one (e.g. DNA integrity or LC3B level in Fig 1B or Fig 3A/B). However, in case of Fig 2 data there are two independent variables (age and mitophagy markers), so the use of two-way ANOVA seems appropriate.

5.) Indication of significant differences seems a bit inconsistent on the figures. (E.g. on Fig 2A: 6m Parkin is significant while 6m PINK1 significance is not indicated. Or on Fig 4: significance of 6m in A is indicated, while the same in B is missing. Fig 3/5/6 is similar.) Please, unify it.

6.) Which ear(s) was used for ABR measurement? The description of ABR measurements are rather sketchy.

7.) The used dose of ketamine (250 mg/kg) and xylazine (116.5 mg/kg) is rather high. 80-100 mg/kg ketamine plus 10-20 mg/kg xylazine is the regular dose. Why did they use so high anaesthetic concentrations?

8.) Fig 1B – What does ‘Mitochondrial DNA integrity’ mean /show? How was it calculated?

Author Response

Responses to the First Reviewer’s Comments

We appreciate the time and effort that the reviewer has dedicated to providing valuable feedback on our manuscript. In respone to reviewer’s comments, we have revised the manuscript as follows.

1) The authors have published a very similar study in the cochlea (Exp Gerontol. 2020 Aug;137:110946). This paper should be cited in this one.

Author’s Response: As per the reviewer’s suggestion, we have cited the article in the Discussion section (page 9, line 223-224) as follows:

“Moreover, reduced mitophagy was observed in aged cochlea, which increased hearing loss [29].”

2) Autophagy may have dual-role (Janus face; e.g. Antioxid Redox Signal 2012, 16:263). It may be favourable to mention and explain this in the Discussion.

Author’s Response: As per the reviewer’s suggestion, we have added the information from the previous report on the dual-role of autophagy in the Discussion section (page 9, lines 189-191) as follows:

“Autophagy may have a dual role. Early upregulation of autophagy may have a pro-survival role and serve to remove damaged mitochondria; however, if this pro-survival attempt fails, autophagy may trigger cell death pathways [20].”

3) The level of auto/mitophagy related RNAs/proteins decreased with age. What could be the cellular location of these decreases? Neurons or Glial cells?

Author’s Response: We investigated the level of mitophagy related RNAs/proteins in the auditory cortex which is composed of tightly packed neurons and glial cells. We think the change occured in both neurons and glial cells. We have added the information in the Results section (page 3, lines 97-98) as follows:

“The effect of aging on mitophagy in the mouse auditory cortex that is composed of tightly packed neurons and glial cells was evaluated by measuring the expression levels of genes and proteins associated with mitophagy in the 1-, 6-, 12-, and 18-month groups.”

4) One-way ANOVA is appropriate for testing the effect of one independent variable (age) on the dependent one (e.g. DNA integrity or LC3B level in Fig 1B or Fig 3A/B). However, in case of Fig 2 data there are two independent variables (age and mitophagy markers), so the use of two-way ANOVA seems appropriate.

Author’s Response: As per the reviewer’s suggestion, we have analyzed the data on age and the expression of mitophagy-related genes and proteins using two-way ANOVA. And, we have incorporated the description in the Methods section 4.6 (page 12, lines 351-353) as follows:

“Comparisons between age groups were analyzed using one-way analysis of variance (ANOVA), followed by the Tukey’s honestly significant difference (HSD) post-hoc test. Data on age and the expression of mitophagy-related genes and proteins were analyzed using two-way ANOVA.”

In addition, we have revised Fig 2.

5) Indication of significant differences seems a bit inconsistent on the figures. (E.g. on Fig 2A: 6m Parkin is significant while 6m PINK1 significance is not indicated. Or on Fig 4: significance of 6m in A is indicated, while the same in B is missing. Fig 3/5/6 is similar.) Please, unify it.

Author’s Response: We have revised the Figs 2A/3/4/5/6 as per the reviewer’s suggestion.

6) Which ear(s) was used for ABR measurement? The description of ABR measurements are rather sketchy.

Author’s Response: The ABR thresholds in both ears were recorded from the scalp of the mice using the computerized Intelligent Hearing System. As per the reviewer’s suggestion, more details have been explained in the Methods section (page 10, lines 254-264) as follows:

“The hearing tests were conducted in a soundproof chamber. The body temperature of mice was maintained at 37°C–38°C by placing the anesthetized mice on a heating pad, and monitored using a rectal probe throughout the recording period. The ABR thresholds in both ears were recorded from the scalp of the mice using the computerized Intelligent Hearing System (IHS, FL, USA) equipped with the Smart-EP software. Subcutaneous needle electrode were inserted on the vertex and overlying the ventral region of the left and right bullae. The tone burst stimuli were generated in waveforms with 256 stimuli (rate: 21.1/s) at frequencies of 8 kHz, 16 kHz, and 32 kHz. The ABR waveforms were recorded on a sampling period of 10 ms from 10–80 dB sound pressure level (SPL) intervals below the maximum amplitude. The ABR threshold was defined as the lowest stimulus level at which response peaks for waves were visible in the evoked trace.”

7) The used dose of ketamine (250 mg/kg) and xylazine (116.5 mg/kg) is rather high. 80-100 mg/kg ketamine plus 10-20 mg/kg xylazine is the regular dose. Why did they use so high anaesthetic concentrations?

Author’s Response: We performed the anesthesia by injecting 70 mg/kg ketamine plus 9 mg/kg xylazine. We have added more details on the dose in the Methods section (page 10, lines 251-254) as follows:

“The mice (n = 15 per group) were anesthetized by injecting ketamine (50 mg/mL, Yuhan, Seoul, Korea) and xylazine (23.32 mg/mL, Bayer Korea, Seoul, Korea). The ketamine/xylazine cocktail was prepared by mixing 3.5 mL of ketamine, 1 mL of xylazine, and 8 mL of sterile water. The cocktail was injected intraperitoneally at a dose of 0.1 mL/20 g bodyweight.”

8) Fig 1B – What does ‘Mitochondrial DNA integrity’ mean /show? How was it calculated?

Author’s Response: We investigated the mtDNA integrity by examining mtDNA damage by calculating the ratio of long mtDNA (10.1 kb, Mus musculus) to short mtDNA (117 bp, Mus musculus) (Wang et al., Stem Cells. 2010, PMID 20954243). We have incorporatd the information in the Methods section (page 11, lines 279-281) as follows:

“The mtDNA integrity was determined by examining mtDNA damage by calculating the ratio of long mtDNA to short mtDNA.”

Reviewer 2 Report

This is an interesting manuscript that provides novel insights into potential mechanisms for age-related hearing loss in the central auditory system in a mouse model. The experiments appear to be designed well and the data produced support the conclusions by the authors.

However the authors did not discuss the potential role of other pathways, including miR-34a/SIRT1, which appears to play an important role in cochlear hair cell loss and hearing deterioration in aging through mitophagy and mitochondrial biogenesis (Xiong H et al. Modulation of miR-34a/SIRT1 signaling protects cochlear hair cells against oxidative stress and delays age-related hearing loss through coordinated regulation of mitophagy and mitochondrial biogenesis. Neurobiology of Aging 79 (2019) 30-42) or Dynamin Related Protein (DRP-1). DRP-1 was reported to initiate mitophagy and eliminate mitochondrial dysfunction in the cochlea. It appears to also protect against oxidative stress-induced senescence. Dynamin 1 may provide a therapeutic target for age-related hearing loss (Lin H. et al, (2019) Inhibition of DRP-1-Dependent Mitophagy Promotes Cochlea Hair Cell Senescence and Exacerbates Age-Related Hearing Loss. Front. Cell. Neurosci. 13:550.doi: 10.3389/fncel.2019.00550). Any therapeutic strategies for ameliorating age-related hearing loss must address the mechanisms involved in the auditory periphery. This group of researchers has reported similar mechanisms for mitophagy in relation to age related hearing loss in the cochlea (Oh J, Youn CK, Jun Y, Jo E-R, Cho S II Reduced mitophagy in the cochlea of aged C57BL/6J mice. Exp Gerontol. 2020 Aug; 137:110946. doi: 10.1016/j.exger.2020.110946. Epub 2020 May 5). The authors should cite this paper in their references.

Author Response

We appreciate the time and effort that the reviewer has dedicated to providing valuable feedback on our manuscript. In respone to reviewer’s comments, we have revised the manuscript as follows.

1) . However the authors did not discuss the potential role of other pathways, including miR-34a/SIRT1, which appears to play an important role in cochlear hair cell loss and hearing deterioration in aging through mitophagy and mitochondrial biogenesis (Xiong H et al. Modulation of miR-34a/SIRT1 signaling protects cochlear hair cells against oxidative stress and delays age-related hearing loss through coordinated regulation of mitophagy and mitochondrial biogenesis. Neurobiology of Aging 79 (2019) 30-42) or Dynamin Related Protein (DRP-1). DRP-1 was reported to initiate mitophagy and eliminate mitochondrial dysfunction in the cochlea. It appears to also protect against oxidative stress-induced senescence. Dynamin 1 may provide a therapeutic target for age-related hearing loss (Lin H. et al, (2019) Inhibition of DRP-1-Dependent Mitophagy Promotes Cochlea Hair Cell Senescence and Exacerbates Age-Related Hearing Loss. Front. Cell. Neurosci. 13:550.doi: 10.3389/fncel.2019.00550). Any therapeutic strategies for ameliorating age-related hearing loss must address the mechanisms involved in the auditory periphery. This group of researchers has reported similar mechanisms for mitophagy in relation to age related hearing loss in the cochlea (Oh J, Youn CK, Jun Y, Jo E-R, Cho S II Reduced mitophagy in the cochlea of aged C57BL/6J mice. Exp Gerontol. 2020 Aug; 137:110946. doi: 10.1016/j.exger.2020.110946. Epub 2020 May 5). The authors should cite this paper in their references.

Author’s Response: As per the reviewer’s suggestion, we have added the information from previous reports on the therapeutic strategies for ameliorating age-related hearing loss and cited the articles in the Discussion section (page 9-10, lines 220- 226) as follows:

“Mitophagy appears to play an important role in cochlear hair cell loss and hearing deterioration during aging. SIRT1 is an NAD-dependent deacetylase that modulates mitophagy. A previous study has shown that activation of SIRT1 increased the levels of PINK1 and Parkin in old mice, and attenuated age-related cochlear hair cell loss [28]. Moreover, reduced mitophagy was observed in aged cochlea, which increased hearing loss [29]. This is supported by a study found that the inhibition of dynamin-related protein-1 (DRP-1), which initiates mitophagy, promotes cochlear hair cell senescence and exacerbates age-related hearing loss [30].”

Round 2

Reviewer 2 Report

The authors have made appropriate responses to reviewer concerns from the previous submission.